# Axial Disease in Psoriatic Arthritis: A Challenging Domain in Clinical Practice

**DOI:** 10.3390/diagnostics14151637

**Published:** 2024-07-30

**Authors:** Lucía Alascio, Ana Belén Azuaga-Piñango, Beatriz Frade-Sosa, Juan C. Sarmiento-Monroy, Andrés Ponce, Sandra Farietta, Jose A. Gómez-Puerta, Raimon Sanmartí, Juan D. Cañete, Julio Ramírez

**Affiliations:** Arthritis Unit, Rheumatology Department, Hospital Clínic Barcelona, Villarroel Street, 170, 08036 Barcelona, Spain; lalascio@recerca.clinic.cat (L.A.); abazuaga@clinic.cat (A.B.A.-P.); frade@clinic.cat (B.F.-S.); sarmiento@clinic.cat (J.C.S.-M.); aponce@clinic.cat (A.P.); farietta@clinic.cat (S.F.); jagomez@clinic.cat (J.A.G.-P.); sanmarti@clinic.cat (R.S.); jcanete@clinic.cat (J.D.C.)

**Keywords:** psoriatic arthritis, spondyloarthritis, biological therapy

## Abstract

Psoriatic arthritis (PsA) is a chronic inflammatory condition affecting about one-third of individuals with psoriasis. Defining axial involvement in PsA (axPsA) remains debated. While rheumatologists guide clinical practice, consensus on axPsA is still lacking. This paper explores historical and upcoming definitions from the Axial Involvement in Psoriatic Arthritis (AXIS) study, which aims to establish a validated axPsA definition. Epidemiological data reveal diverse axPsA prevalence rates, emphasizing its complex relationship with peripheral arthritis and enthesitis. Unique genetic, clinical, and radiological features differentiate axPsA from ankylosing spondylitis (AS), necessitating refined classification criteria. The recommendations from the Assessment of Spondylarthritis international Society (ASAS) provide valuable guidance due to the limited direct evidence. Emerging therapies, including interleukin-23 (IL-23) inhibitors or Janus kinase (JAK) inhibitors, are under investigation for axPsA. Currently, secukinumab, an interleukin-17 (IL-17) inhibitor, is an evidence-based option for axPsA management. However, given the variability in individual patient responses and disease manifestations, personalized, evidence-based treatment approaches remain essential for optimizing patient outcomes. In the final section, two real-life cases illustrate the challenges in managing axPsA, emphasizing the importance of tailored therapies. Achieving precision in defining axPsA remains a formidable task, making detailed criteria essential for effective strategies and improving patient outcomes.

## 1. Introduction

Psoriatic arthritis (PsA) is a chronic inflammatory disease that typically manifests in the context of psoriasis (Pso), affecting up to one-third of Pso patients, with the highest incidence occurring approximately ten years after the onset of cutaneous involvement [1,2]. PsA can affect multiple domains [3,4], with the axial skeleton being an area currently generating significant debate due to its similarities and differences with ankylosing spondylitis (AS) [5,6], as well as its controversial response to novel therapeutic targets such as IL-23. In this review, we aimed to look at the concept, epidemiology, clinical characteristics, diagnosis, and treatment of axial PsA (axPsA), with a special focus on clinical responses to IL-17 and IL-23 inhibitors. To show the complexity of this domain in clinical practice, we finally present two clinical cases with unexpected onset of axial symptoms in patients with PsA while in treatment with IL-17 inhibitors.

## 2. Concept

Various terms have been used in the medical literature to describe the axial component of PsA, including psoriatic spondylitis, psoriatic spondyloarthritis, or the more recently accepted term, axial PsA (axPsA) [7,8,9]. Currently, there is no consensus on the definition of axPsA. In clinical practice, the rheumatologist’s opinion prevails since there are no specific diagnostic/classification criteria for axial involvement in PsA [10,11,12]. For research, both the Assessment of Spondylarthritis international Society (ASAS) criteria for axial spondyloarthritis (axSpA) and the Classification Criteria for Psoriatic Arthritis (CASPAR) criteria for PsA can be used to classify axPsA patients [13]. However, the ASAS criteria are more stringent and may select a subset of axPsA patients, potentially failing to capture the full heterogeneity of the axial dimension in PsA. In contrast, the CASPAR criteria are more inclusive and closely resemble what a rheumatologist considers axPsA in clinical practice, making them more suitable for research studies focusing on this aspect of PsA.

Part of the ongoing debate about whether axPsA represents a subgroup of axSpA or a distinct entity is primarily due to the uncertainty regarding the effectiveness of IL-23 inhibitors in the treatment of axPsA [14,15,16].

There are not many studies in the literature that have specifically targeted axPsA patients as their population of interest. In these studies, the definitions used for axPsA are diverse and heterogeneous (Table 1). In the MAXIMISE study (Managing AXIal Manifestations in psorIatic arthritis with SEcukinumab) aimed to evaluate secukinumab efficacy in axPsA patients [17], inclusion criteria were based solely on the clinical judgement of rheumatologists. Although imaging studies were used for patient assessment, they were not required for inclusion. This inclusive approach ensured that patients without imaging evidence of axPsA were not excluded.

In the DISCOVER study, which assessed the safety and efficacy of guselkumab for active PsA, a “mixed opinion” approach was used to combine clinical judgement and imaging findings [18]. Clinical criteria involved the rheumatologist’s assessment of symptoms and disease progression, while imaging evidence, such as radiology or MRI, was also considered. This dual approach aimed to include a broader range of axPsA patients, including those without clear radiographic changes but with clinical signs of axial involvement. In other analyses, such as post hoc studies of upadacitinib and ustekinumab in PsA, only clinical criteria were used for selecting axPsA patients [19,20].

In contrast, studies from the University of Toronto Psoriatic Arthritis Clinic used more stringent criteria. These studies often selected patients who met the modified New York criteria for AS. The New York criteria include both clinical and radiographic standards, such as chronic back pain and stiffness lasting more than 3 months, and radiographic evidence of sacroiliitis on X-ray imaging [21]. This rigorous approach ensures a more precise diagnosis of AS, differentiating it from other forms of inflammatory arthritis.

Finally, studies conducted using data from the COREVITAS cohort (registry from the United States) used clinical or imaging criteria indistinctly [22,23]. Therefore, it could be summarized that the common denominator in these studies is the presence of Pso, with the rheumatologist’s opinion typically being the first additional criterion, and, in some cases, imaging findings being added.

The Axial Involvement in Psoriatic Arthritis (AXIS) study is a collaborative initiative between the ASAS and the Group for Research and Assessment of Psoriasis and Psoriatic Arthritis (GRAPPA) groups, with the primary goal of establishing a valid definition of axPsA for research studies [24]. While in clinical practice, the rheumatologist’s opinion will likely remain the gold standard for diagnosing axPsA (similar to axSpA), this new definition from the AXIS study will allow us to select homogeneous groups of patients for epidemiological, research studies, and clinical trials. Specifically, the AXIS study aims to analyze the frequency of axial involvement in PsA, identify inflammatory and structural changes observed through imaging in these patients, and try to identify clinical, genetic, and laboratory factors associated with axPsA. As classification criteria for patient inclusion, the CASPAR criteria were employed, selecting patients with less than 10 years of disease duration to avoid cases with extensive structural damage or degenerative changes. Exclusion criteria included contraindications for imaging studies, such as conventional radiology or magnetic resonance imaging (MRI), and prior use of systemic therapies due to their potential impact on structural damage. In any case, the results of the AXIS study are eagerly anticipated as they will enable us to conduct research studies with homogeneous populations. This will enhance our understanding of the characteristics of the axPsA patient population.

**Table 1 diagnostics-14-01637-t001:** Variability in classification of axial PsA across studies.

Study	Inclusion Criteria
MAXIMIZE [17] Clinical trial of secukinumab involving 498 patients with axPsA who had an insufficient response to NSAIDs.	Clinical criteria: PsA diagnosis determined by CASPAR criteria, BASDAI > 4, and spinal pain VAS > 40.
DISCOVER-1 and 2 [18] Post hoc exploratory analysis of guselkumab clinical trials involving 381 patients with active PsA in DISCOVER-1 and 739 patients in DISCOVER-2.	Clinical and imaging criteria: Active PsA (DISCOVER-1: ≥3 swollen joints, ≥3 tender joints, CRP ≥0.3 mg/dL despite standard therapies; DISCOVER-2: ≥5 swollen joints, ≥5 tender joints, CRP ≥ 0.6 mg/dL despite standard therapies) with current or past sacroiliitis on imaging as assessed by the local investigator.
Axial Disease in Psoriatic Arthritis Study [25]. Single-center study involving 201 patients with PsA and 201 patients with AS.	Clinical and imaging criteria: PsA diagnosis classified by CASPAR criteria, with a diagnosis of Pso (past or present), and radiographic sacroiliitis as per the modified New York criteria for AS.
Aydin et al. [26]. Registry study of 1186 patients with PsA (PsART).	Clinical criteria: Inflammatory back pain present; no specific imaging criteria needed.
Queiro and Cañete [27]. Medical records review of 70 patients with psoriasis and radiographic evidence of SpA.	Clinical criteria: Defined according to ASAS classification criteria for axSpA.

AS: ankylosing spondylitis; ASAS: Assessment of Spondyloarthritis international Society; axSpA: axial spondyloarthritis; BASDAI: Bath Ankylosing Spondylitis Disease Activity Index; CASPAR: Classification of Psoriatic Arthritis; CRP: C-reactive protein; NSAID: non-steroidal anti-inflammatory drug; PsA: psoriatic arthritis; PsART: Psoriatic Arthritis of Turkey; Pso: psoriasis; SpA: spondyloarthritis; VAS: visual analogue scale.

## 3. Epidemiology

Due to the different definitions used for axPsA over time, the prevalence of this domain is not well established. An initial study by a Canadian group reported that up to 34% of patients diagnosed with PsA exhibited axial involvement [28]. A subsequent study by Torre Alonso et al. found that 23% of 180 PsA patients demonstrated axial involvement [29]. This Spanish study introduced several interesting concepts. On the one hand, it suggested that axial involvement in PsA could affect not only the sacroiliac joints but also the spine exclusively, a concept not observed in axSpA, where there is typically an upward progression from sacroiliac joints to the cervical spine. On the other hand, it acknowledged that most axPsA patients exhibit mixed involvement, with other dimensions such as peripheral arthritis or enthesitis also playing a significant role. This aspect may be relevant when determining treatment strategies since, in less than 5%, the axial domain is the only manifestation of the disease. Lastly, the prevalence of the HLA-B27 gene differed between patients with bilateral sacroiliitis (with higher prevalence, within the ranges observed in patients with AS) and those with unilateral sacroiliitis, where only 22% were HLA-B27 positive [30]. Therefore, the presence of the HLA-B27 gene may influence the disease phenotype, although it does not appear to impact the therapeutic response, as demonstrated in an analysis of the North American COREVITAS registry, where therapeutic response remained independent of the presence of the HLA-B27 gene [31].

In any case, isolated axial involvement in PsA does not seem to be common, as evidenced by a recent study where only 2% of over 1500 PsA patients exhibited exclusively spinal involvement [32]. In contrast, nearly 30% of this patient population demonstrated mixed involvement, a combination of axial and peripheral involvement. Pso, however, was not frequently observed in AS patients, where fewer than 5% of nearly 1700 patients had skin involvement. Furthermore, when comparing these two patient cohorts, it was observed that axPsA patients were older than those with AS and had milder symptoms, as indicated by a lower prevalence of inflammatory axial pain.

## 4. Clinical and Radiological Characteristics

Experience from clinical practice shows distinct patterns of disease in patients with axPsA and patients with axSpA without Pso, although overlap is inevitable in a subgroup of patients. In a recent study conducted in Bath, United Kingdom, the authors compared the clinical, genetic, and radiological characteristics of three patient groups: peripheral psoriatic arthritis (pPsA), axPsA, and AS [33]. It was found that up to 25% of axPsA patients exhibited silent axial disease, defined as radiographic changes without clinical symptoms. Furthermore, 24% of axPsA patients met the modified New York criteria for AS, and one-third displayed spinal involvement without concurrent sacroiliac joint involvement. Notably, axPsA patients had a lower prevalence of HLA-B27 when compared to AS patients, and they also had a lower burden of structural damage, both in the spine and sacroiliac joints.

In terms of radiological characteristics, axPsA shares similarities with AS but also exhibits notable differences [9,25,34]. AS patients more frequently manifest bilateral and symmetrical sacroiliitis, with syndesmophytes and ankylosis being more common. In contrast, axPsA patients tend to present with unilateral sacroiliitis, and when bilateral, it often displays an asymmetric pattern. Structural damage is less severe and less frequently observed in axPsA patients, with a lower incidence of ankylosis. AxPsA patients may also present with isolated vertebral involvement, sometimes accompanied by minimal or absent sacroiliitis. Additionally, the presence of enthesitis and peripheral arthritis is more frequently observed in axPsA. Thus, while there is overlap between AS and axPsA, they also exhibit distinct features that set them apart (Table 2).

Consequently, based on clinical, genetic, and radiological data, axPsA may be considered a distinct entity from AS [35,36].

## 5. Axial Psoriatic Arthritis Cohorts in Clinical Practice

Over recent years, multiple studies analyzing the clinical characteristics of axPsA patients from several cohorts worldwide (Table 3) have been published. In a recent study conducted in Germany, the authors investigated the incidence of axPsA in a Pso cohort of patients under 45 years who had a history of axial pain for more than three months [37]. Following clinical assessments by rheumatologists and imaging evaluations using conventional radiology and MRI, fifteen out of one hundred patients were diagnosed with axPsA based on the rheumatologist’s criteria, with an additional five patients being diagnosed with PsA. Among these fifteen axPsA patients, fourteen met the CASPAR criteria, while only nine met the ASAS criteria, indicating that the CASPAR criteria align more closely with the clinical opinion of the rheumatologist. Among the axPsA patients, 28% tested positive for HLA-B27, and their serum C-reactive protein (CRP) levels were significantly higher than patients with Pso. All axPsA patients exhibited imaging changes, with 36% showing spinal changes without sacroiliac joint involvement and one-third meeting the New York criteria for AS.

In the Swiss registry, more than 8000 patients were analyzed, including 5800 with axSpA, 10% of whom had Pso, and over 2500 PsA patients, of whom up to 43% had axial involvement [38]. All patients exhibited equivalent disease activity, physical function, and spinal mobility, underscoring that both entities (axSpA and axPsA) share the same disease burden. In the analysis of the German Rabbit registry, more than 2800 patients were selected [39]. Among the 1400 axSpA patients, 12% had Pso, while 25% of the 1400 PsA patients had axial involvement. Within the latter group, 45% of patients were diagnosed with axPsA based only on clinical criteria, 17% based only on imaging (without clinical symptoms), and up to 40% of patients met both clinical and imaging criteria. Overall, among patients diagnosed with axPsA by the rheumatologist’s criteria, 82.7% and 54.5% had clinical and imaging involvement, respectively.

In the analysis of the Spanish REGISPONSER registry, patients with AS and Pso (constituting 10% of the total) were compared to patients with axPsA [40]. Patients with AS and Pso exhibited a higher prevalence of the HLA-B27 gene, inflammatory axial pain, uveitis, and a greater burden of structural damage. Finally, in the analysis of the North American COREVITAS registry, which included over 1500 patients, patients with axPsA were compared to those with peripheral PsA [23,31]. The most noteworthy finding was that patients with axial involvement scored worse on scales of pain, fatigue, and quality of life, highlighting the increased disease burden from the patient’s perspective.

**Table 3 diagnostics-14-01637-t003:** Summary of observational studies on axial involvement in psoriatic arthritis.

Reference	Country	Study Objective
Gladman DD et al. (1987) [28]	Canada	To analyze the clinical characteristics and patterns of PsA in 220 patients.
Alonso JCT et al. (1991) [29]	Spain	To perform a clinical, immunological, and radiological study of 189 patients with PsA.
Queiro R et al. (2002) [30]	Spain	To compare clinical features between HLA-B27-positive and HLA-B27-negative psoriatic spondyloarthropathy.
Mease PJ et al. (2018) [23]	USA	To analyze the influence of axial involvement on clinical characteristics of PsA using data from the Corrona Psoriatic Arthritis/Spondyloarthritis Registry (axial versus peripheral PsA).
Mease PJ et al. (2022) [31]	USA	To assess treatment responses in PsA patients with axial disease based on HLA-B27 status using the CorEvitas Psoriatic Arthritis/Spondyloarthritis Registry.
Kwok TSH et al. (2022) [32]	Canada	To investigate isolated axial disease in PsA and ankylosing spondylitis with Pso.
Proft F et al. (2022) [37]	Germany/International	To identify early axial PsA among psoriasis patients through a prospective multicenter study.
Ciurea A et al. (2023) [38]	Switzerland	To characterize patients with axial PsA and axial spondyloarthritis with concomitant psoriasis using data from the SCQM registry.
Regierer AC et al. (2023) [39]	Germany	To compare patients with axial PsA and axial spondyloarthritis with concomitant psoriasis using the RABBIT-SpA register.
Michelena X et al. (2022) [40]	Spain	To characterize the axial phenotype of PsA and compare it with ankylosing spondylitis with psoriasis using data from the REGISPONSER registry.

HLA: human leukocyte antigen; PsA: psoriatic arthritis; Pso: psoriasis; RABBIT-SpA: Rheumatoid Arthritis: Observation of Biologic Therapy—Spondyloarthritis; REGISPONSER: Registry of Spondyloarthropathies in Spain; SCQM: Swiss Clinical Quality Management.

## 6. Therapeutic Approach

Given the limited evidence regarding the effectiveness of treatments for axPsA, the current recommendations advise adhering to the axSpA guidelines [41,42,43]. Therefore, when evaluating a patient with PsA and axial symptoms, it is recommended as a first step to practice physical exercise alongside non-steroidal anti-inflammatory drugs (NSAIDs). Similar to axSpA, the recommended approach involves using at least two NSAIDs at maximum doses for 6–8 weeks, followed by an evaluation of the treatment response. If therapeutic goals are not met, the initiation of biological treatment or Janus kinase (JAK) inhibitors is recommended.

In assessing axPsA patients, it is crucial to note that, currently, all effective therapeutic options for axSpA can be applied to axPsA. However, there are uncertainties regarding the efficacy of IL-23 inhibitors, given the negative findings in AS studies. Both ASDAS-CRP and BASDAI are validated for axSpA but not specifically for axPsA [44,45]. Many axPsA patients also have peripheral symptoms like arthritis, enthesitis, or dactylitis, which can influence these disease activity measures. Thus, these tools should be used cautiously.

As previously mentioned, there is no direct evidence of the efficacy of any treatment in axPsA coming from clinical trials, except for secukinumab, which has demonstrated its efficacy in the MAXIMISE study [17]. TNF inhibitors were only evaluated in a longitudinal observational study, reaching a BASDAI50 response in 31.2% of patients at 12 months [9]. Further insights are provided by post hoc analyses of the PSUMMIT and SELECT-PsA trials for ustekinumab and upadacitinib, respectively [19], and the DISCOVER trial subanalysis for guselkumab [18]. In total, 31% of patients in SELECT-PsA 1 and 35.7% in SELECT-PsA 2 had axial involvement by investigator judgment. Greater proportions of patients achieved BASDAI50 with UPA15 versus placebo [46]. In the pooled PSUMMIT-1&2 of ustekinumab, the physician-reported spondylitis (*n* = 223) subset was analyzed. Mean Week 24 changes were larger among ustekinumab than placebo-treated patients for both neck/back/hip pain (−1.99 vs. −0.18) and mBASDAI (−2.09 vs. −0.59) [47]. In the DISCOVER subanalysis, one-third of participants with both axial and peripheral involvement showed better BASDAI responses with guselkumab compared to placebo, suggesting the potential utility of IL-23 inhibitors in such patients. However, this hypothesis requires validation in the ongoing STAR trial [48].

The efficacy of IL-23 inhibitors in axPsA has faced criticism, particularly regarding the validity of BASDAI and ASDAS-CRP for patients without confirmed axial involvement [49]. Enhancement in these parameters may be due to peripheral improvement or central sensitization. Another issue is why IL-23 inhibitors do not work in AS [14]. Studies have shown that the IL-23/IL-17 pathway is not linear, with significant IL-17 production independent of IL-23, particularly from innate lymphoid cells [50,51]. Some resident entheseal cells may have IL-23 receptors, while others may not. Additionally, it cannot be ruled out that sensitivity to IL-23 may change over time, with only IL-23 blockade being effective in the early stages of axial disease, as suggested by an animal model from Baeten et al. [52]. Another theory is that the entheses are anatomically different in peripheral sites, where the synovial-entheseal complex is present, encompassing subcutaneous cellular tissue, perientheseal fat, bursa, or fibrocartilage, and in axial sites, where the predominant feature is bone edema without inflammation of surrounding tissues. Based on these anatomical differences, McGonagle D et al. hypothesized a main bone involvement in AS, where inflammation of bony anchorage of the enthesis and MRI osteitis prevail, while in axPsA, a phenotype closer to age-related changes and diffuse idiopathic skeletal hyperostosis appears, with a soft tissue centric involvement. These differences could justify a different therapeutic approach for both diseases. Moreover, in AS, HLA-B27+ status is linked to diffuse spinal osteitis, and key inflammatory processes take place in the red marrow. The florid osteitis at enthesis anchorage sites is a myeloid-lineage-rich compartment with neutrophil and macrophage lineage cells. Perhaps that could explain why the standard dose of IL-23 inhibitors is ineffective because of the inability to fully neutralize myeloid-lineage-derived IL-23. On the contrary, in axPsA, there is a comparative lack of osteitis; PsA involvement is soft-tissue-centric, similar to the peripheral skeleton, where the synovio-entheseal complex soft tissues are the key target of inflammation. Ligament tissue is predominantly composed of extracellular matrix with the majority of cells of fibroblastic lineage with a comparative paucity of immune cells, although cells of myeloid and lymphoid lineage are present. That could explain why secondary outcomes from pivotal trials suggest that IL-23 inhibition is efficacious in PsA [53].

Whether axPsA and AS are the same entity or different diseases remains an open question. A small proof-of-concept study by Fitzgerald et al. showed greater improvement in axPsA patients compared to AS and mechanical axial pain patients after intramuscular corticosteroid treatment [54]. A post hoc analysis comparing axPsA patients from the DISCOVER trials of guselkumab with AS patients from ustekinumab trials found differences in genetic and serologic profiles, suggesting potential distinctions between the diseases [55]. In this indirect analysis, patients were found to be different both genetically, with a higher prevalence of the HLA-B27 gene in AS, and serologically, with significantly higher serum levels of IL-17A and IL-17F in axPsA patients compared to AS patients. While these studies are not definitive due to their design, they raise the hypothesis that axPsA and AS may differ in certain aspects, potentially influencing treatment responses.

The MAXIMISE study is the only clinical trial demonstrating the efficacy of a molecule in axPsA, with axial involvement as the primary endpoint. Secukinumab showed a significantly better clinical response compared to placebo at 12 weeks. Criticism of the trial includes the use of clinical criteria without imaging for inclusion, aiming to capture the full spectrum of axPsA, including patients without imaging confirmation. Despite these criticisms, the clinical response rates were comparable to those in AS trials, with significant decreases in bone edema observed [17,39].

## 7. Challenging Management Paradigms in Psoriatic Arthritis: A Clinical Perspective

Despite secukinumab, an IL-17 inhibitor, being the only drug with evidence-based efficacy in axPsA, unexpected situations can be observed in clinical practice. We will now describe two clinical cases of patients who developed new symptoms of axial disease while undergoing therapy with an IL-17 inhibitor. Both cases were confirmed with imaging evidence of inflammation and demonstrated clinical and imaging improvement following a change in therapy.

### 7.1. Case Study 1

A 58-year-old Caucasian male with a history of smoking, obesity, hypercholesterolemia, and cutaneous Pso was diagnosed with PsA due to dactylitis affecting both hands and feet, along with concomitant axial involvement manifested as inflammatory low back pain. Subsequent radiographic assessments revealed asymmetric bilateral sacroiliitis, leading to the initiation of secukinumab at a dosage of 150 mg every 4 weeks in combination with methotrexate (MTX) at a dose of 15 mg weekly. Persistent dactylitis and cutaneous manifestations necessitated an increase in the secukinumab dosage to 300 mg every 4 weeks. This adjusted therapy regimen led to rapid clinical remission across the joint (axial and peripheral) and cutaneous domains.

One year after achieving remission, the patient presented with nocturnal pain in the dorsal spine, an area that had not been previously affected. He required NSAIDs nightly (naproxen 500 mg every 12 h) for symptom relief. Suspecting an axial flare, an MRI was performed, revealing bone marrow edema in the anterior portion of the D6 vertebral body (Figure 1). Elevated C-reactive protein levels also indicated an axial flare in a previously unaffected region. Secukinumab was stopped, and infliximab was initiated at a dose of 5 mg/kg every 8 weeks, resulting in significant clinical improvement in the nocturnal pain. Subsequent MRI scans confirmed the resolution of bone marrow edema.

### 7.2. Case Study 2

A 60-year-old female with a history of smoking, hypertension, and psoriasis presented with dactylitis and arthritis of the left carpal joint. Initial diagnostic tests, including rheumatoid factor and anti-cyclic citrullinated peptide (anti-CCP) antibody tests, were negative, ruling out rheumatoid arthritis. Corticosteroid joint infiltrations partially alleviated joint symptoms initially.

Given the incomplete relief from initial treatments, MTX therapy was initiated at a dose of 15 mg weekly to target both joint and cutaneous symptoms. This dose was maintained consistently throughout the treatment period. Persistent dactylitis prompted the addition of etanercept (administered at a dose of 50 mg weekly) to the treatment regimen, which was continued for two and a half years to better manage the articular symptoms. When this combination failed to achieve the desired outcomes, namely reduction in pain and improvement in joint function, the treatment was switched to adalimumab (40 mg every other week). However, this transition was associated with worsening joint symptoms characterized by increased pain, swelling, reduced range of motion, and other clinical signs of inflammation, as well as a reactivation of cutaneous manifestations.

Recognizing the need for a different approach, ixekizumab was introduced at a dose of 160 mg initially, followed by 80 mg every 4 weeks as maintenance therapy, resulting in marked improvement across both cutaneous and articular domains. Approximately one year after having reached remission with ixekizumab, the patient experienced incapacitating lower back pain. MRI scans revealed significant bone edema in the lumbar vertebral bodies (Figure 2), indicating new axial involvement.

Despite a lack of prior axial symptoms, which had been confirmed by normal sacroiliac joint radiographs and a negative HLA-B27 test at disease onset, the patient experienced this unexpected event while in treatment with ixekizumab. This new manifestation presented complex challenges in management, as joint and skin domains were well-controlled at the time of the axial flare.

Ixekizumab was discontinued, and infliximab therapy was initiated to effectively manage the emerging axial reactivation. Infliximab was administered as an initial dose of 5 mg/kg, followed by 5 mg/kg at 2 and 6 weeks, and subsequently every 8 weeks. Three months into the infliximab treatment, the patient’s lower back pain significantly improved, while peripheral and cutaneous remission was maintained. Follow-up MRI demonstrated complete resolution of bone marrow edema from the lumbar vertebral bodies (Figure 2).

The clinical experiences in both patients highlight the complex nature of managing PsA, particularly in terms of possible unexpected responses to IL-17 inhibitors. Loss of efficacy over time (secondary failure), the need for higher doses for spine involvement, or paradoxical reactions associated with IL-17 inhibition similar to lupus-like syndrome or psoriasis after TNF inhibition [56,57] arise as possible explanations for these unexpected events. Moreover, when prescribing IL-17 inhibitors, caution is advised for patients with clinical or subclinical inflammatory bowel disease (IBD), as these inhibitors may exacerbate IBD, requiring careful patient selection and monitoring [58].

In any case, these clinical cases stress the importance of being vigilant about varied treatment responses due to the intricate relationship between different aspects of the disease. As the paradigm of PsA therapy continues to evolve, these examples emphasize the significance of continuous monitoring and the imperative for personalized, evidence-based therapeutic interventions. Additionally, the emerging concept of unexpected events with IL-17 inhibitors underscores the need for more research to understand the underlying causes and develop targeted strategies to address these challenges.

Consequently, these findings emphasize the urgent need for effective therapeutic strategies tailored specifically for axPsA, building on insights from studies like the MAXIMISE trial, which demonstrated the efficacy of secukinumab in managing this challenging aspect of PsA. As research progresses and our understanding of the molecular mechanisms underlying axPsA grows, the pursuit of targeted therapeutic approaches that address the nuances of this condition remains a critical priority.

## 8. Conclusions

Defining axPsA remains a subject of ongoing debate. In clinical practice, the subjective judgement of the rheumatologist continues to be the gold standard for diagnosis. For research, we look forward to the outcomes of the AXIS study—a collaborative initiative by the ASAS and GRAPPA groups—aimed at establishing a suitable definition. This will enable the recruitment of more homogeneous patient populations and promote a common language in research.

From both clinical and imaging perspectives, patients with axPsA share characteristics with those having axSpA. Nevertheless, they exhibit distinctive features that set them apart as a distinct entity, potentially transcending the confines of being considered merely a subgroup of axSpA. The question of whether these distinctions have therapeutic significance remains unanswered.

While the presence of HLA-B27 is relevant in defining specific disease phenotypes, it does not appear to influence therapeutic response. However, we reflect on the possibility that patients with axPsA may require higher doses of biological treatments compared to those without this condition. This suggestion emerges as a reflection warranting further investigation to better understand the underlying pathophysiological mechanisms that could explain this difference.

It is crucial to recognize that most patients with axSpA also have substantial involvement in other domains, such as peripheral arthritis or enthesitis. Therefore, treatment decisions for each patient should be based on a comprehensive assessment that considers not only the axial component of the disease but also all domains affecting patients’ quality of life.

## Figures and Tables

**Figure 1 diagnostics-14-01637-f001:**
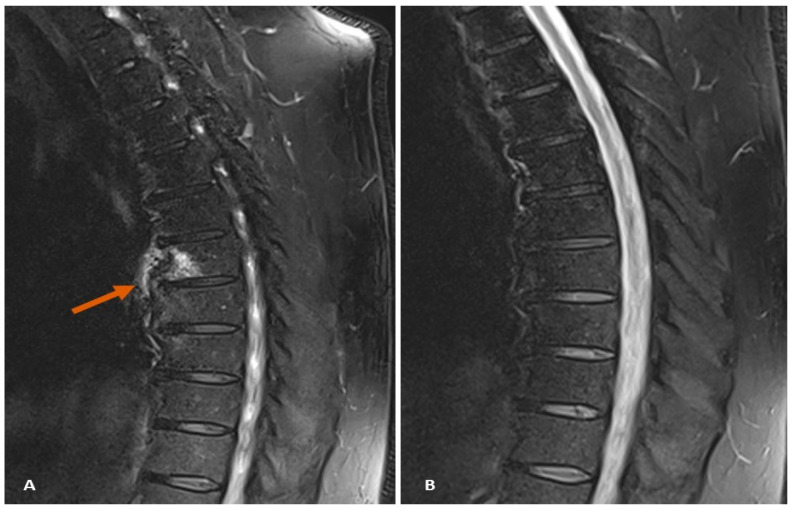
Dorsal spine MRI in STIR sequence. (**A**) Bone marrow edema at T6 vertebral corner (arrow). (**B**) Resolution of bone edema after 3 months of infliximab treatment.

**Figure 2 diagnostics-14-01637-f002:**
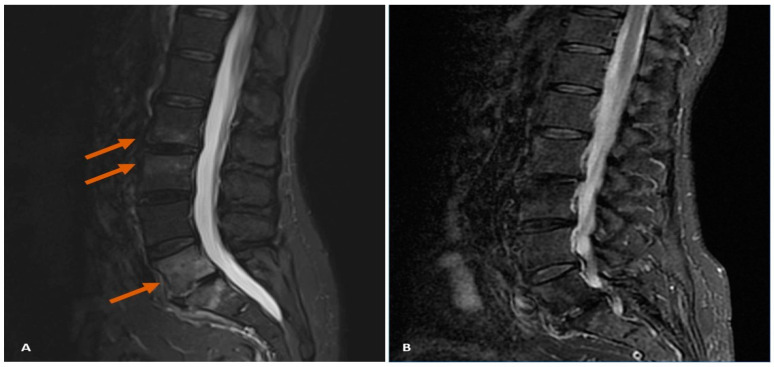
(**A**) Bone edema in the lumbar vertebral bodies (marked by orange arrows). (**B**) Complete resolution of bone marrow edema after 3 months of treatment with infliximab.

**Table 2 diagnostics-14-01637-t002:** Feature comparison of axPsA and AS.

Feature	axPsA	AS
New York criteria for AS	24% of patients meet the criteria	All patients
Spinal involvement	Earlier neck involvement is observed. May present with isolated vertebral involvement	Always associated with sacroiliitis
HLA-B27 prevalence	Lower compared to AS (30–40%)	Higher prevalence (>90%)
Structural damage	Lower burden, both in spine and sacroiliac joints	Higher burden
Sacroiliitis	Often unilateral, usually asymmetric if bilateral	Frequently bilateral and symmetrical
Syndesmophytes and ankylosis	Less common	More common
Enthesis and peripheral arthritis	More frequently observed	Less emphasis

axPsA: axial psoriatic arthritis; AS: ankylosing spondylitis; HLA-B27: human leukocyte antigen B27.

## Data Availability

The data presented in this study are available upon request from the corresponding author.

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
