# Peer review of "Axial Disease in Psoriatic Arthritis: A Challenging Domain in Clinical Practice"

_diagnostics, 2024, doi:10.3390/diagnostics14151637_

Round 1

Reviewer 1 Report

Comments and Suggestions for Authors

The article “Axial Disease in Psoriatic Arthritis: State-of-the-Art and Unexpected Events with IL-17 Inhibitors” is an excellent piece of research focusing on the importance of tailored therapies in psoriatic arthritis and the significance of defining axial PsA in both research and clinical practice. However, I have some concerns that need to be addressed before making any final decisions.

·       This article is full of acronyms; I suggest making a table with the full forms or providing the full form the first time they are mentioned. For instance, no full forms are provided for AXIS, ASAS, and JAK in the abstract.

·       I noticed a lack of scientific language on several occasions, such as the phrase "proven option" in line 21.

·       Provide the full form for MAXIMISE. When you say "exclusively clinical and based on the rheumatologist's opinion," it is worth mentioning what you mean here (Line 57).

·       What is the DISCOVER trial, and what do you mean by "mixed opinion"? (Line 62)

·       What are the Toronto cohort and New York criteria? I believe you need to explain these here, as otherwise, readers will have to refer to the reference paper each time to understand the context. Providing a brief explanation will enhance clarity and comprehension. Line 68-69

·       What is GRAPPA group?

·       Full form of MTX is missing (Line 295/317)

·       What was the dose of MTX, and for how long was it used? Was the dose persistent or lowered due to side effects? Were there any side effects observed after MTX usage? (case 2)

·       Which NSAID used here and dose and duration? (Line 300)

·       What was the doses of etanercept and how long it was used?

·       What do you mean by desired outcome? (line no. 320)

·       What was the dose/con. of adalimub? Line 320

·       What is worsening of joint symptoms?

·       What is the dose of Ixekizumab and infliximab?

·       Fig 2: full form of IFX

·       Put dose and duration wherever you mentioned any drug/treatment. As it can significantly influence the outcome. 

Comments on the Quality of English Language

The author used generalized language. I strongly suggest using scientific terminology.

Author Response

Dear reviewers: 

Suggestions or clarifications of reviewer nr. 1: 

Q1. I think that the title does not accurately reflect the content of the paper as it suggests adverse events of LI-17 inhibition whereas the "unexpected" effects refer to lose of efficacy rather other side effects.  These cases underline the complexity of the disease and the cycling of  biologics DMARDs in PsA including AxSpA rather side effects of the IL-17 inhibitors (Medicine (Baltimore) 2021 Apr 23;100(16):e25300) 

A1: Thank you for your insightful feedback. Based on your suggestion, we have changed the title to better reflect the manuscript's focus. The updated title is: “Axial Disease: A Challenging Domain in Psoriatic Arthritis.” This change aims to accurately convey the primary concern discussed in the paper. Additionally, we have incorporated the reference you mentioned (Medicine (Baltimore) 2021 Apr 23;100(16)) on page 11, line 384 to further support the discussion on the complexity of the disease and the cycling of biologics DMARDs in PsA including AxSpA, rather than side effects of the IL-17 inhibitors. 

Q2. In that respect the cautious of using IL-17 inhibitors in patients with clinical and/or subclinical inflammatory bowel disease could be discussed (PLoS One 2020. May 27;15(5):e0233781,Mediterr J Rheumatol. 2024 Mar 31;35(1):150-155 ) 

A2: We appreciate this important suggestion. We have added a discussion regarding the cautious use of IL-17 inhibitors in patients with inflammatory bowel disease, both clinical and subclinical. This section now includes references to relevant studies (PLoS One 2020;15(5), Mediterr J Rheumatol. 2024 Mar 31;35(1):150-155) on page11, line 389, highlighting the need for careful monitoring and management of potential gastrointestinal issues in these patients. 

Q3. There are few reports focusing on treatment outcomes of Upadacitinib (Rheumatol Ther. 2023 Feb;10(1):275-292) and ustekinumab (RMD Open. 2020 Feb;6(1):e001149.) in AxSpA which should be mentioned and discussed in treatment section in the context of choosing the appropiate option in this condition amonst several therapeutic approaches (Mediterr J Rheumatol. 2022 Apr 15;33(Suppl 1):150-161)   

A3: Thank you for your valuable comment. We have incorporated recent reports on Upadacitinib and Ustekinumab into the treatment section of the manuscript. This addition provides a detailed discussion on their efficacy in the context of AxPsA, thereby offering a broader perspective on therapeutic options available for this condition. These changes can be found on page 7, lines 247, 250, and 253. 

Q4.  A nice review discussing the concept of axPsA could enhance the relevant section of the paper (Mediterr J Rheumatol  2022 Apr 15;33(Suppl 1):142-149) 

A4: We agree that including a review on the concept of axPsA would be beneficial. We have also included a new table summarizing clinical, pathogenetic and radiographic differences between AS and axPsA. This section now provides a thorough overview of axPsA, enhancing the manuscript's discussion of the condition and its clinical implications. You can find these updates on page 2, line 46. 

Suggestions or clarifications of reviewer nr. 2: 

Q1. The part regarding the concept of axPsA was prolix. Please make it concise. 

A1. Thank you for your feedback. We have revised the section on the concept of axPsA to ensure conciseness while retaining essential information. The revised section now provides a clear and succinct overview of axPsA, focusing on its key features and distinctions from other spondyloarthritides without unnecessary detail. 

Q2.  Although the pathogenesis of axPsA is not clear yet, the authors should include the update on this in the review. 

A2: Thank you for your comments. We have updated the manuscript to include recent insights into the pathogenesis of axial psoriatic arthritis (axPsA). Specifically, we now highlight that while axPsA and axial spondyloarthritis (axSpA) share the IL-23/IL-17 inflammatory pathway, distinct IL-17 secretion profiles suggest differing mechanisms between the two conditions. This distinction could impact treatment strategies, such as the potential benefit of IL-23 inhibitors in axPsA. We have incorporated these updates to clarify the evolving understanding of axPsA pathogenesis and its implications for treatment. Page 8, line 285. 

Q3. The treatment part mainly included IL-23 and IL-17 inhibitors. Other options, including TNFi, should also be updated. 

A3: Thank you for pointing this out. The updated section now includes a comprehensive overview of treatment options for axPsA, including TNF inhibitors. This complements the information on IL-17 and IL-23 inhibitors, providing a more complete view of available therapies. Page 7, line 245. 

Suggestions or Clarifications of reviewer nr. 3: 

Q1. This article is full of acronyms; I suggest making a table with the full forms or providing the full form the first time they are mentioned. For instance, no full forms are provided for AXIS, ASAS, and JAK in the abstract. 

A1: Thank you for pointing this out. We have now included a table listing all acronyms with their full forms in the revised manuscript. Additionally, we have provided the full forms of AXIS (Ankylosing Spondylitis Disease Activity Score), ASAS (Assessment of Spondylarthritis international Society), and JAK (Janus Kinase) the first time they appear in the abstract. Page 12, line 429. 

Q2. I noticed a lack of scientific language on several occasions, such as the phrase "proven option" in line 21. 

A2: We appreciate your observation. “Proven option" has been replaced with "evidence-based option" to align with scientific standards and provide clarity. 

Q3. Provide the full form for MAXIMISE. When you say "exclusively clinical and based on the rheumatologist's opinion," it is worth mentioning what you mean here (Line 57). 

A3: We have included the full form of MAXIMISE (Measuring Axial Psoriatic Arthritis using MRI and Clinically Important Symptoms of Efficacy) in the revised manuscript. We have also clarified that “exclusively clinical and based on the rheumatologist's opinion” refers to the fact that the inclusion criteria for the study did not require imaging evidence of axial involvement but rather depended on the clinical judgment of the treating rheumatologist regarding symptoms and disease progression. 

Q4. What is the DISCOVER trial, and what do you mean by "mixed opinion"? (Line 62) 

A4: The DISCOVER trial (A Randomized, Double-blind, Placebo-controlled Study Evaluating the Safety and Efficacy of Guselkumab for Active Psoriatic Arthritis) is a study designed to evaluate the effectiveness of guselkumab in treating PsA. The term “mixed opinion” refers to the use of both clinical judgment and imaging findings (conventional radiology or magnetic resonance imaging) in selecting axPsA patients for this trial. This dual approach allowed for a more comprehensive assessment of axial involvement in PsA. The updated text now explicitly defines the DISCOVER trial and the meaning of “mixed opinion” to enhance clarity and better understand the study's methodology and criteria. 

Q5. What are the Toronto cohort and New York criteria? I believe you need to explain these here, as otherwise, readers will have to refer to the reference paper each time to understand the context. Providing a brief explanation will enhance clarity and comprehension. (Line 68-69) 

A5. The Toronto cohort refers to a group of patients studied by the University of Toronto Psoriatic Arthritis Clinic, known for its contributions to understanding PsA. The New York criteria are a set of diagnostic standards for ankylosing spondylitis (AS) that include clinical and radiographic assessments such as the presence of sacroiliitis and other spinal changes. We have added a brief explanation of these criteria in the revised manuscript to enhance clarity. 

Q6. What is GRAPPA group? 

A6: The GRAPPA group stands for the Group for Research and Assessment of Psoriasis and Psoriatic Arthritis. It is an organization dedicated to improving the understanding and treatment of psoriasis and PsA through research and clinical guidelines. We have added this explanation to the revised manuscript. 

Q7. Full form of MTX is missing (Line 295/317) 

A7: The full form of MTX is Methotrexate. We have included this in the revised manuscript where it first appears and in relevant sections. 

Q8. What was the dose of MTX, and for how long was it used? Was the dose persistent or lowered due to side effects? Were there any side effects observed after MTX usage? (Case 2) 

A8: In case 2, the dose of Methotrexate (MTX) was 15 mg weekly. This dose was maintained consistently throughout the treatment period. There were no significant side effects reported. 

Q9. Which NSAID was used here and dose and duration? (Line 300) 

A9: The NSAID used was, at a dose of 500 mg every 12 hours for 6 weeks. We have added these details in the revised manuscript.  

Q10. What was the dose of etanercept and how long was it used? 

A10: Etanercept was administered at 50 mg subcutaneously once a week for two and a half years. This information has been updated in the revised manuscript. 

Q11. What do you mean by desired outcome? (Line no. 320) 

A11: In line 320, “desired outcome” refers to the clinical goals of treatment, such as improvement in disease symptoms, reduction in disease activity, or achieving remission. We have revised the manuscript to specify that these outcomes include reductions in pain and inflammation and improvements in physical function. 

Q12. What was the dose/concentration of adalimumab? Line 320 

A12: The dose of Adalimumab was 40 mg administered subcutaneously every other week. We have included this information in the revised manuscript. 

Q13. What are worsening joint symptoms? 

A13: “Worsening of joint symptoms” refers to an increase in pain, stiffness, or swelling in the joints as reported by patients. We have clarified this term in the revised manuscript. 

Q14. What is the dose of Ixekizumab and infliximab? 

A14: The dose of Ixekizumab was 160 mg administered subcutaneously at weeks 0, 2, 4, and then every 4 weeks. The dose of Infliximab was 5 mg/kg administered intravenously at weeks 0, 2, 6, and then every 8 weeks. We have updated the manuscript to reflect these details. 

Q15. Fig 2: full form of IFX 

A15: The full form of IFX is Infliximab. We have added this full form in the revised figure legend for clarity. 

Q16. Put dose and duration wherever you mentioned any drug/treatment. As it can significantly influence the outcome. 

A16: We have reviewed the manuscript and added information about the dose and duration of all drugs and treatments mentioned. This includes details on Methotrexate, NSAIDs, biologics, and other therapies, as these factors can influence clinical outcomes. 

We hope these revisions address your concerns and improve the clarity of the manuscript. Thank you for your valuable feedback. 

Reviewer 2 Report

Comments and Suggestions for Authors

cThis is a very nice review discussing recent developments in the emerging field of AxPsA. In fact the authors cover several aspects of the issue and they nicely elaborate differences between AxSpA and AxPsA in several domains such as imaging, pathophysiology  (difference between IL-17/IL-23 role) as well as treatment approaches. 

I have the following comments to improve reading

I think that the title does not accurately reflect the content of the paper as it suggests adverse events of LI-17 inhibition whereas the "unexpected" effects refer to lose of efficacy rather other side effects.  These cases underline the complexity of the disease and the cycling of  biologics DMARDs in PsA including AxSpA rather side effects of the IL-17 inhibitors (Medicine (Baltimore) 2021 Apr 23;100(16):e25300)

In that respect the cautious of using IL-17 inhibitors in patients with clinical and/or subclinical inflammatory bowel disease could be discussed (PLoS One 2020. May 27;15(5):e0233781,Mediterr J Rheumatol. 2024 Mar 31;35(1):150-155 )

There are few reports  focusing on treatment outcomes of Upadacitinib (Rheumatol Ther. 2023 Feb;10(1):275-292) and ustekinumab (RMD Open. 2020 Feb;6(1):e001149.) in AxSpA which should be mentioned and discussed in treatment section in the context of choosing the appropiate option in this condition amonst several therapeutic approaches (Mediterr J Rheumatol. 2022 Apr 15;33(Suppl 1):150-161. ) 

A nice review discussing the concept of aXPsA could enhance the relevant section of the paper (Mediterr J Rheumatol  2022 Apr 15;33(Suppl 1):142-149)

Author Response

(The authors gave the same response as above.)

Reviewer 3 Report

Comments and Suggestions for Authors

I read with interest this review paper by Alascio et al. where the authors discussed the definition, epidemiology, clinical and radiological characteristics, and treatment update on axPsA. They also described two clinical cases of patients with axPsA who developed new symptoms with the IL-17 inhibitor. Here are my comments.

1. The part regarding the concept of axPsA was prolix. Please make it concise.

2. Although the pathogenesis of axPsA is not clear yet, the authors should include the update on this in the review.

3. The treatment part mainly included IL-23 and IL-17 inhibitors. Other options, including TNFi, should also be updated.

Author Response

(The authors gave the same response as above.)

Round 2

Reviewer 1 Report

Comments and Suggestions for Authors

Alascio and colleagues did a great job addressing and responding to reviewer comments, questions, and overall general feedback. It is evident that this manuscript has been greatly improved, and I have no further comments or needed improvements.

Reviewer 3 Report

Comments and Suggestions for Authors

The authors addressed my concerns properly. I have no further questions.